# Human Psychophysiology Is Influenced by Low-Level Magnetic Fields: Solar Activity as the Cause

Michael Hanzelka [1,*], Jiří Dan [2], Pavel Fiala [1] and Přemysl Dohnal [1]

1   Department of Theoretical and Experimental Electrical Engineering, Brno University of Technology, 616 00 Brno, Czech Republic; fialap@feec.vutbr.cz (P.F.); dohnalp@vutbr.cz (P.D.)
2   Department of Psychology, Faculty of Arts and Letter, Catholic University in Ružomberok, 034 01 Ružomberok, Slovakia; jiri.dan@ku.sk
*   Correspondence: xphanzelka@vutbr.cz; Tel.: +420-54114-6280

**Abstract:** We evaluate the impact of changes in solar activity on three significant human psychophysiological parameters: skin conductance, electromyography (EMG), and the share of abdominal and diaphragmatic breathing in overall ventilation. Variations affecting human psychophysiology due to changes in solar activity directly document the assertion that psychology, behavior, and decision-making all reflect geomagnetic field alterations that stem from variable solar activity. The relevant experiments showed that solar processes, during which the Earth is exposed to electrically charged particles from the Sun (solar wind), exert an impact on the psychophysiological parameters of the body.

**Keywords:** BioGraph Infiniti; skin resistance; abdominal/diaphragmatic breathing; EMG





## 1. Introduction

This article outlines the data and knowledge obtained from experiments centered on the low-level magnetic fields that emerge through solar activity variations and affect geomagnetic stability [1]. The concept and interpretations of the research are closely described in relevant references [2–6]. In general terms, the interdisciplinary research examines those psychophysiological and other measurable human parameters which may change due to solar activity-induced processes in low-level (low-frequency) magnetic fields. The subdisciplines and problems involved range within, above all, physics, medicine, electro- and geomagnetism, atmospheric science, and cosmic meteorology [1].

Regarding low-level magnetic fields, these comprise low-frequency fields at $f = 0.01$–3 kHz within the following bands: the ultra low frequency (ULF) region, $f_u = 300$ Hz–3 kHz; the super low frequency (SLF) region, $f_{slf} = 30$–300 Hz; and the extremely low frequency (ELF) region, $f_{el} = 0.1$–30 Hz [7].

The above frequency bands are enshrined in relevant regulations issued by the International Commission on Non-Ionizing Radiation Protection (ICNIRP) [8]; in these documents, however, the boundary values of the amplitudes are markedly higher, such as the 50 mT in $B_{lim}$. The value defines the maximum change of the magnetic flux within a field having the frequency of $f = 1$ Hz, assuming the presence of persons in this type of environment. Problematically, however, the $B_{lim}$ boundary value multiple times exceeds the values measured to date in atmospheric magnetic field changes arising from solar eruptions and their consequences, i.e., geomagnetic storms [9].

In the presented context, we discuss the impacts of low-level fields induced by solar activity variations on some of the parameters characterizing or found in a healthy individual, these parameters being skin resistance, $R_s$; muscular contractions, $D_s$; and the proportion of thoracic breathing to diaphragmatic respiration, $O_{t-d}$.

The evaluation of the experiments as set out in [2,3] consisted in determining whether and, possibly, to what extent solar activity variations affect the measurable parameters of the mental and psychophysiological stability of the human body.

Importantly, the conclusions made after the completion of the individual research phases have been related to the assumed or proved impacts of solar activity variations on human health, exploiting an analysis of human behavior in stress situations; such moments induce behavior at which we can observe changes in the measurable and/or quantifiable psychophysiological parameters of the human body, including human body parameters such as the heart rate or breathing frequencies, skin resistance, body temperature, and muscular contractions. In the research specified within [4–6], the stress on the participants was generated by using the Stroop color test [10] and a mathematical task (subtracting the number 7 from 1071), the evaluated factors being the overall correctness and the time to achieve the result. Each of these stress states, $t_z$, lasted two minutes and had been designed to simulate the standard load on a human organism during regular daily life situations. The total time to measure a respondent equaled $T_{sumr}$ = 19 min; the procedure comprised three rest (or relaxation) stages, each lasting $T_{rel}$ = 5 min, and two stress phases, $t_z$, separated from one another by the rest ones. Expectably, the experiments showed that individuals with higher levels of psychophysiological lability are more prone to reflecting variation in the intensity of solar activities, such variation manifesting itself especially through altered skin resistance and respiration frequency, together with more frequent transitions from diaphragmatic breathing into the thoracic type. Another notable effect that arises from the psychophysiological shift rests in more frequent muscular contractions in labile persons [4–6].

The total numbers of respondents related to the count of stress load measurements are indicated in Table 1. The eventual set of participants was comparatively narrow, as we had intended the recruitment to yield a homogeneous sample of persons aged 18 to 25. These individuals, all affiliated with the Brno-based University of Defence, had to pass demanding entrance tests to fit our purposes; the testing focused on physical and mental resilience. The value of 210 in Table 1 expresses the total number of measurement cycles.

**Table 1.** The counts of measurements completed and respondents involved [3].

| Respondents | Measurements |
|:---:|:---:|
| 4 | 1 |
| 1 | 2 |
| 4 | 3 |
| 12 | 4 |
| 24 | 5 |
| 4 | 6 |
| 49 | 210 |

The outcomes of the experiment, which started on 22 April 2014 and continued until 26 June 2014, were eventually correlated with the solar activity intensity data measured by NASA and NOAA in the same period [11]. During the two months, we relied on NASA-produced TXT files capturing the activity between 1874 and the year of the research. In these files, the daily collected details referred especially to the number of sunspots. At present, solar wind data can be shared via, for instance, the website https://omniweb.gsfc.nasa.gov/ (accessed on 30 November 2021).

After completing the above input stages, we compared the stress load and human behavior-related information in the social context, targeting a different set of subjects from within the general population; the actual comparison was performed by utilizing socioeconomic indexes such as the DJIA [12] and the S&P 500 [13]. From the perspective of our research, the indexes represent human economic behavior (and its social dimension) on capital markets, where the decision-making involves high stress loads and responsibility rates. Interestingly, the parameter processing results reflect variations in the intensity of

solar activity. The risk-based behavior of an investor can be characterized as emotional, influenced by investment psychosis. Overall, capital markets are considered a zone where emotions prevail over pragmatism [14]. In psychophysiological terms, emotions embody a measurable indicator, especially as regards evaluating the patterns and alterations in skin resistance, $R_s$. The measurability of emotions finds application also in other fields, such as data verification and polygraph interrogations [15]. Further, skin resistance variations are evaluated to provide psychophysiological biofeedback [16], which facilitates practicing human concentration skills and allows sensing responses to external stimuli that induce emotional processes; the stimuli and responses are, most often, visual or audial.

The experiment and its transposition into the social sphere enabled us to search responses to the following crucial questions:

(A)  Is human psychophysiology influenced by changes in solar activity?
(B)  Do solar activity variations embody a major factor affecting human mental states, behavior, and decision-making?

To consider and analyze these issues, we employed the methodology below.

## 2. Materials and Methods

The concepts and progress of the experiments to measure low-level electromagnetic fields impacting human psychophysiology in consequence of altered solar activity have already been described in detail [2,3]; some significant information, however, still remains to be presented.

The data processing exploited above all correlation analysis, whose primary aim was to confirm the following hypotheses:

**Hypotheses 1 (H1).** *Low-level magnetic fields generated through solar activity exert a negative impact on human beings, influencing their behavior and decision-making.*

**Hypotheses 2 (H2).** *There is a significant interaction of the human low-level magnetic and electromagnetic fields on the one hand and geomagnetic variation-induced low-level magnetic fields on the other.*

**Hypotheses 3 (H3).** *Solar activity, the resulting geomagnetic storms, and economic behavior and decision-making are directly interrelated.*

During the experimentation, solar activity was being recorded by NASA, which collects applicable daily measurement values; a record thereof is visualized in Figure 1 below.

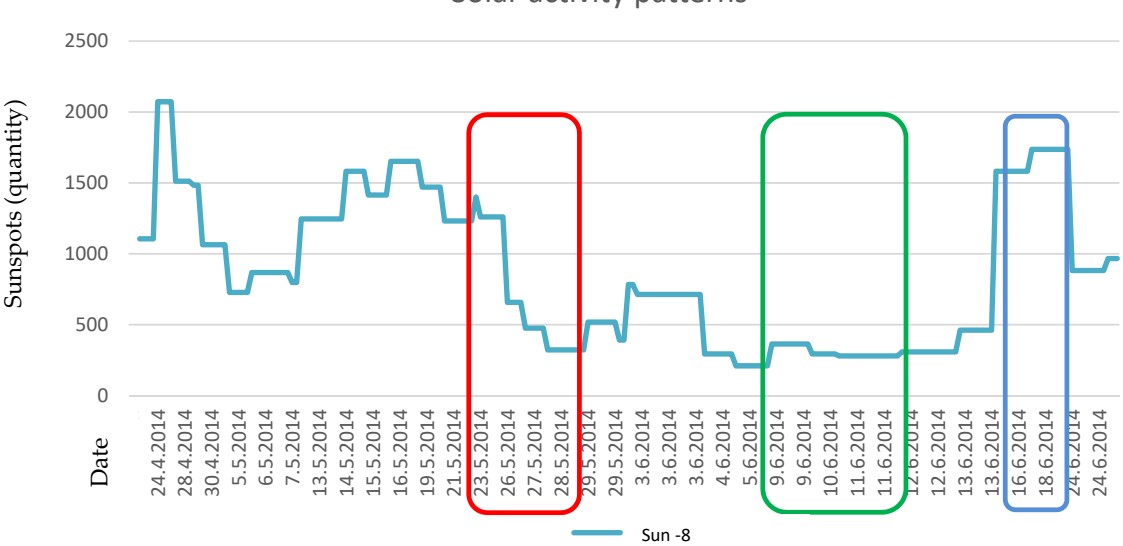

**Figure 1.** The solar activity during the research.

The parameter "Sun–8 days" in Figure 1 characterizes the impact of solar wind particles, which, depending on the wind speed, impinge on the Earth's magnetosphere with a delay of 2 to 8 days; the variations (high/low) and top values are highlighted in diverse colors. According to [17], solar wind can be interpreted as precipitation-free plasma moving at a velocity $v_\mathrm{p}$ = 300–800 km/s in a radial direction, away from the Sun. The wind's concentration (particle density) changes within the range $\rho_\mathrm{p}$ = 1–10 cm$^{-3}$, and the thermal energy scope corresponds to $W_\mathrm{p}$ = 1–30 eV. The mass spectrum is dominated by protons (96%) and—albeit markedly less so—helium nuclei (4%); the share of the "heavier" components is, for the purposes of the research, negligible.

The experiments relied on supporting psychological tests that were carried out via the ASS-SYM [18] method, which determines in the respondents the following states:

- SYM_I, physical and mental exhaustion;
- SYM_II, nervousness and mental stress;
- SYM_III, psychophysiological disorder;
- SYM_IV, performance and behavior disorder;
- SYM_V, pain burden;
- SYM_VI, self-identification and self-control issues;
- ASS_SYM, general symptoms and problems.

The outcomes of the procedure are specified below. The actual measurement was executed with a BioGraph Infiniti device from the Canadian manufacturer Thought Technology Ltd. (https://thoughttechnology.com/) (accessed on 30 November 2021).

## 3. Results and Discussion

The image in Figure 2 displays the correlation between a change in solar activity and the above-outlined mental states identified in the participants via the ASS-SYM. The graphical elements indicate that a more intense solar activity is accompanied by dominating SYM II or SYM_IV, while the opposite condition somewhat surprisingly features a set composed from factors SYM_I, SYM IV, and SYM V. To define how the mental and psychophysiological parameters correlate with solar activity variations, we employed the Pearson correlation coefficient (1):

$$r = \frac{\sum (x - x')(y - y')}{\sqrt{\sqrt{\sum (x - x')^2}(y - y')^2}},\tag{1}$$

where $r$ denotes the coefficient, and $x$ and $y$ are the mean values of selection.

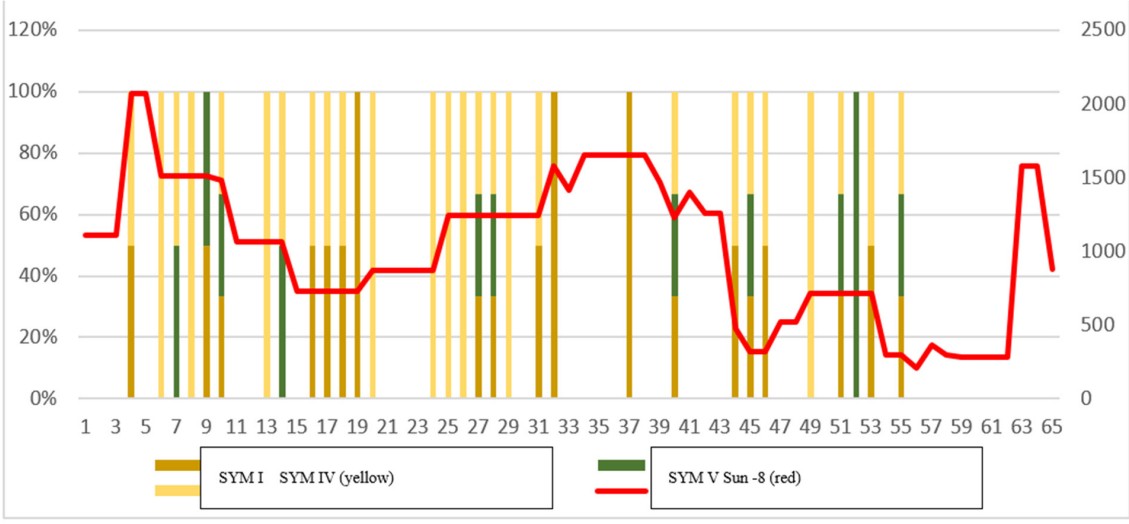

**Figure 2.** Solar activity correlated with the states SYM_I, SYM_IV, and SYM_V.

The patterns in Figure 3 represent the correlations established in the most sensitive psychophysiological quantity, i.e., skin resistance ($R_s$); the indicator "E: skin conductance mean (uS)—actual skin conductance" evaluates the relationship between the conductance and solar activity variation. The correlation diagram confirms the elementary hypothesis (H0) in that the magnetic component of the plasma impinges on the Earth's magnetosphere (and is responded to) with an average delay of 8 days; at this exact point of −8 days, the skin conductance and the intensity of solar activity correlate to the maximum degree.

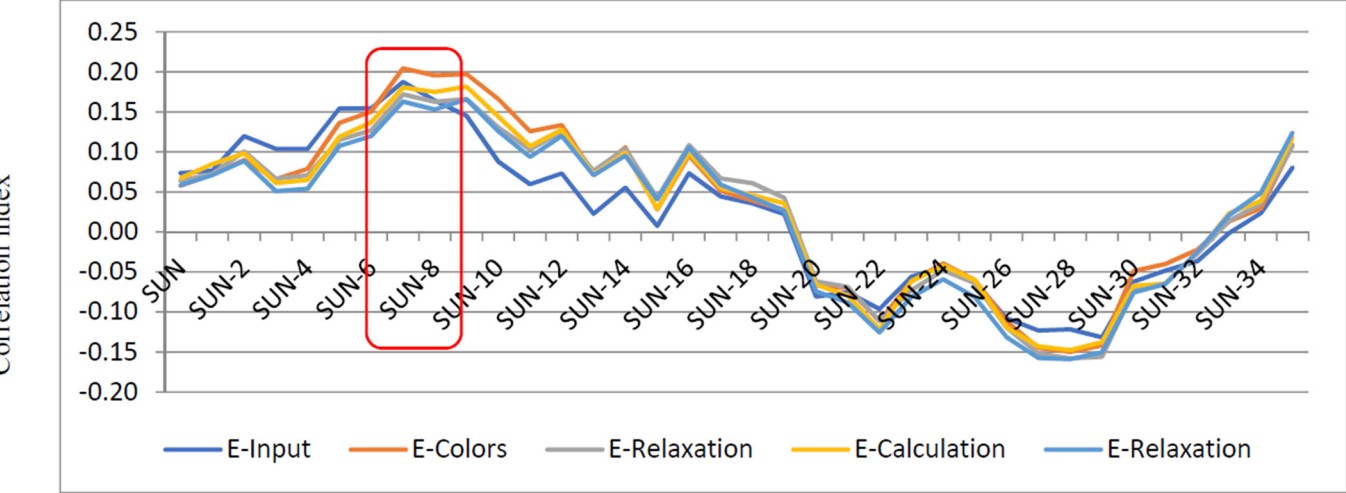

**Figure 3.** The correlation between the psychophysiological quantity "E: skin conductance mean (uS)-actual skin conductance" and the intensity of solar activity.

The relationships of the psychophysiological quantity $D_m$, "D: EMG mean (µV)", are visualized in Figure 4; this quantity represents facial muscle contractions. At rest, the muscles exhibit a contraction value that is expressed through the voltage $u_c$ = 0.0–5.0 µV [19]. By extension, the indicator denotes the level of the overall muscular tension in the organism, the correlation between such tension and the intensity of solar activity being again the highest at approximately −8 days.

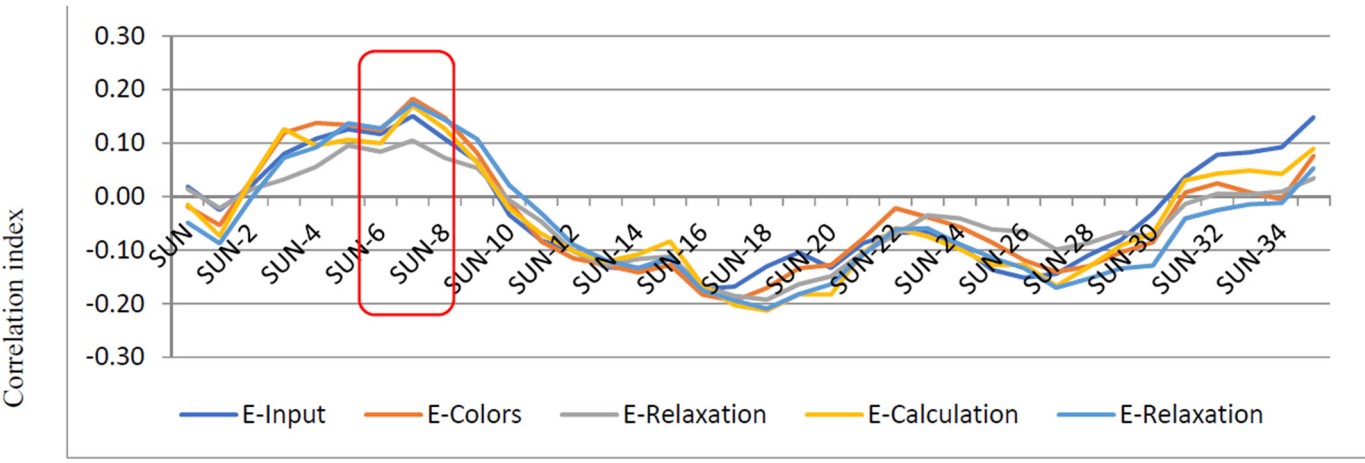

**Figure 4.** The correlation between the psychophysiological quantity "D: EMG mean (uV)-facial muscle contractions" and the intensity of solar activity.

The patterns in Figure 5 convey the relationships of the psychophysiological quantity $O_{t-d}$ "G: abd amplitude mean", namely the share of abdominal (diaphragmatic) breathing in the total pulmonary ventilation. In this ventilation, a normal condition is assumed to

be a state when abdominal breathing prevails. The image then shows that at −8 days the share of diaphragmatic breathing decreases due to altered solar activity.

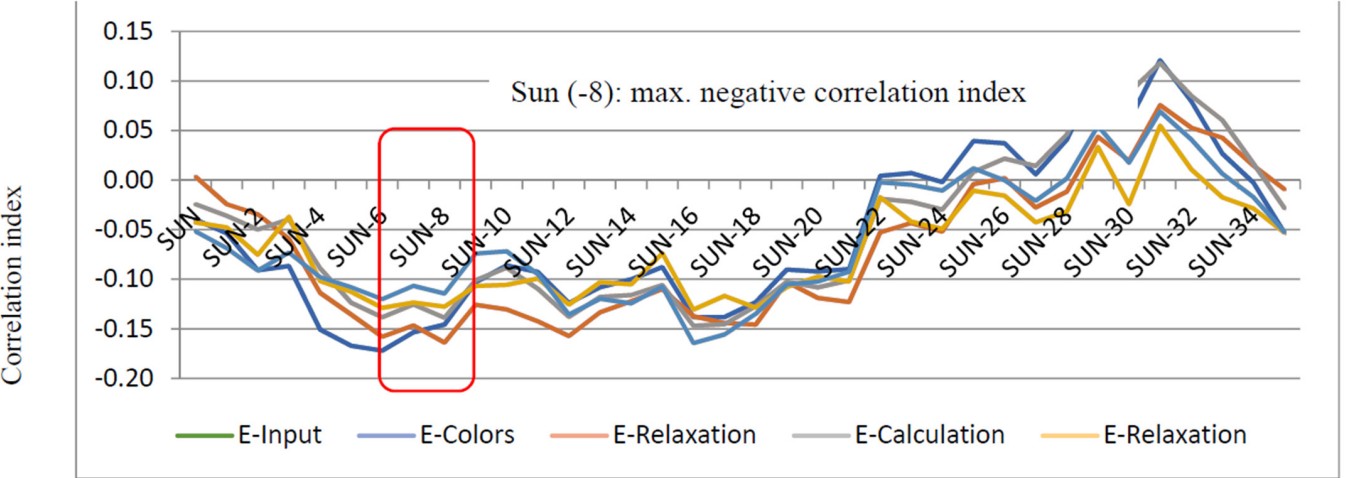

**Figure 5.** The correlation between the psychophysiological quantity "G: abd amplitude mean (rel)-abdominal breathing share" and the intensity of solar activity.

The conclusions arising from the research of the impact exerted by solar activity variations on human psychophysiology can be generalized as follows: We may positively claim that psychophysiological changes in a human being are prominently caused by the discussed alterations (assuming also other, somewhat less significant parameters in the given context). The relevant processes then affect human behavior in terms of the ability to manage the stress produced by everyday life activities and tasks. The measurable impact was observed via correlating the Dow Jones Industrial Average index with shifts in the intensity of solar activity (Figure 6). The monitoring period fully corresponded to the time of the actual laboratory experimentation (24 April 2014 to 26 June 2014). The pattern in Figure 6 expresses a short-term relationship between the DJIA and the intensity of solar activity. At the point of −8 days there appears the first negative correlation maximum, indicating that the index dropped concurrently with a growth in solar activity. The second and third maxima, at −18 and −32 days, respectively, denote the overall impact exerted by an altered intensity of solar activity on the DJIA; the variation and the index, as a matter of fact, embody the cause and the result, affecting the complex system of socioeconomic and interpersonal relationships. Such a condition suggests that, initially, the cause (solar activity variation) acts on the human psychophysiological parameters, and the result (solar change) occurs only in consequence of this process; in our case, the result lies in the dropping DJIA, an index that reflects concrete human psychologies on the capital market.

The 24th solar cycle (Figure 7) began and ended with the solar minima of 2008, 2019, and 2020. Based on problem-related analyses of historical data from NASA records [20], capital market slumps were anticipated, meaning also drops in the DJIA and S&P 500 indexes, among others. This forecast originally concerned the final months of 2019 and the first ones of 2020, proving its reliability later (Figure 8) [13]. Analogously, the shifts in the intensity of solar activity predicted by the National Oceanic and Atmospheric Administration (NOAA) [11] for the last third of 2019 are included in Table 2. The data in the table expose a solar activity decrease down to a minimum at the end of the eleven-year-long cycle. Solar activity was expectable, considering the number of coronal holes and also the character of the electromagnetic effects, namely the frequencies and wavelengths of the measured signals (the radio flux at 10.7 cm and its reflection abilities in the Earth's ionosphere).

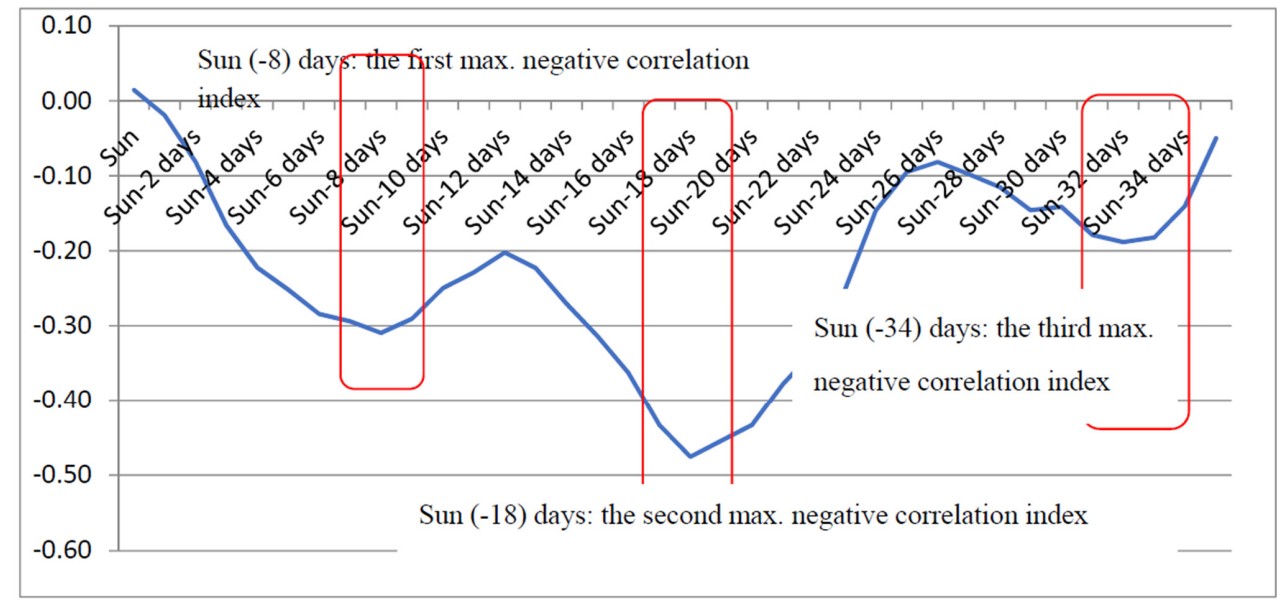

**Figure 6.** The long-term correlations between the DJIA stock market index and the intensity of solar activity.

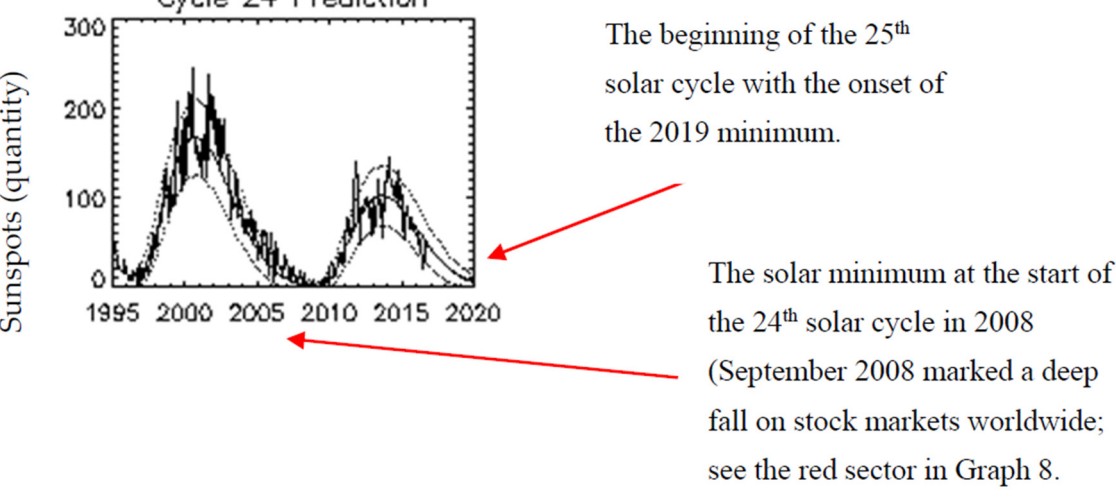

**Figure 7.** The prediction of the 24th cycle. Source: [20].

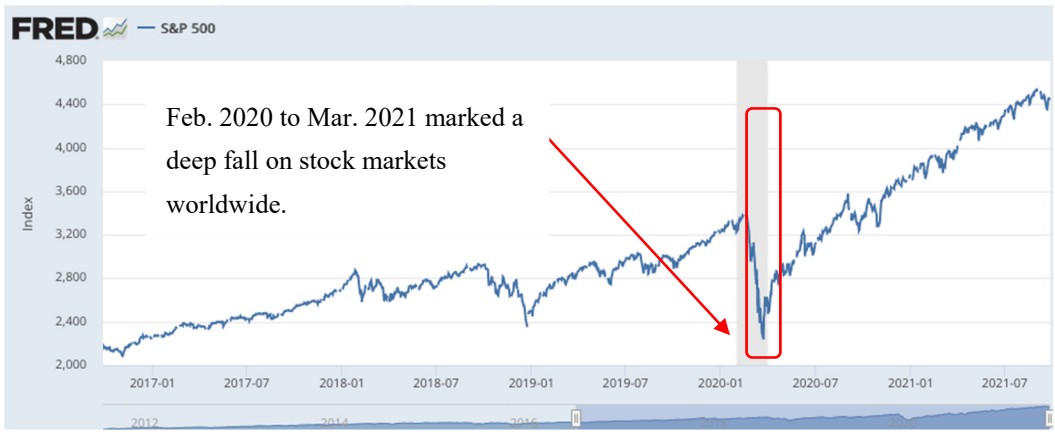

**Figure 8.** The S&P 500 index. Source: [13] FRED (https://fred.stlouisfed.org/series/SP500) (accessed on 30 November 2021).

**Table 2.** The predicted numbers of sunspots and radio flux values with expected ranges (2019). Source: [11].

| | Sunspot Number | | | 10.7 cm Radio Flux | | |
|---|---|---|---|---|---|---|
| **YR MO** | **PREDICTED** | **HIGH** | **LOW** | **PREDICTED** | **HIGH** | **LOW** |
| September 2019 | 5.1 | 15.1 | 0.0 | 63.4 | 72.4 | 60.0 |
| October 2019 | 4.8 | 14.8 | 0.0 | 63.1 | 72.1 | 60.0 |
| November 2019 | 4.4 | 14.4 | 0.0 | 62.8 | 71.8 | 60.0 |
| December 2019 | 4.1 | 14.1 | 0.0 | 62.5 | 71.5 | 60.0 |

The data in Table 2 originate from desk research performed in 2019; these details have nevertheless ceased to be available online. Reference [11], by extension, outlines the prediction for the years 2021 and 2022.

The outcomes of the experiments demonstrate that solar activity variations affect multiple psychophysiological criteria and effects, including skin resistance, muscular contractions, and the proportion of diaphragmatic respiration to thoracic breathing. Importantly, the variations also alter the human mental state, influencing the behavior and decision-making in diverse essential fields, such as the investment and trading domains.

## 4. Conclusions

In the context of this article, the primary experimental research of low-level electromagnetic fields showed that geomagnetic field alterations generated by solar eruptions affect numerous psychophysiological factors. A deeper insight into the role and importance of such changes enables human beings to integrate themselves socially and to interact successfully with other individuals while the overall impact on the negative personal and environmental parameters is minimized. Such parameters include, above all, mental, behavioral, and decision-making shifts that are measurable by means of relevant psychophysiological indicators interpreted in relation to the variability of low-level magnetic fields. The observed field variations were demonstrably induced by geomagnetic processes in the environment, or, by another definition, geomagnetic storms arising from altered solar activity. The parameters and solar wind intensity were subjected to comprehensive evaluation.

As regards the preformulated hypotheses, we reached the following conclusions:

The hypothesis H0, "low-level magnetic fields generated through solar activity exert a negative impact on human beings, influencing their behavior and decision-making", was **confirmed via the applied method**.

The hypothesis H1, "there is a significant interaction of the human low-level magnetic and electromagnetic fields on the one hand and geomagnetic variation-induced low-level magnetic fields on the other", remains to be expanded and supported with further investigation, especially in terms of human brain waves and their interaction with the above-outlined external factors.

The hypothesis H2, "solar activity, the resulting geomagnetic storms, and economic behavior and decision-making are directly interrelated", was also **confirmed via the applied method**.

Follow-up experiments are planned to sense human EEG activities and to determine how they correlate with low-level electromagnetic fields.

The knowledge and skills acquired during the project are of fundamental significance for risk predictability and elimination within multiple fields and subsectors, such as social science, economics, marketing, medicine, transportation, and various industries.

**Author Contributions:** M.H., 50%: experimental psychophysiological research and evaluation of the obtained data; J.D., 25%: experimental psychological research and evaluation of the obtained data; P.F. and P.D., 25%: experimental research support, organization of the respondents, and methodological assistance. All authors have read and agreed to the published version of the manuscript.

**Funding:** This research was funded by [Grant Agency of the Czech Republic] grant number [GA 17-00607S].

**Institutional Review Board Statement:** The Ethics Council did not have to approve the research because the information/data entering the research had been anonymized and all of the respondents had submitted the required Informed Consent document.

**Informed Consent Statement:** The anonymized research data are not public and remain stored in servers of Brno University of Technology, Department of Theoretical and Experimental Electrical Engineering.

**Data Availability Statement:** Some of the research outputs are publicly available from https://www.vutbr.cz/en/students/final-thesis/detail/104368 (accessed on 29 November 2021).

**Acknowledgments:** The research was funded via National Sustainability Program, grant No. LO1401, and supported within a grant of Czech Science Foundation (GA 17-00607S). For the actual analyses and experiments, the current infrastructure of the SIX Center was used. The authors acknowledge the help and support of the University of Defence (Brno), whose students and employees formed a homogeneous sample of respondents to facilitate the investigation procedures.

**Conflicts of Interest:** The authors declare no conflict of interest relating to this article. The sponsors did not influence in any manner the collection, analysis, and interpretation of the data, and they were not involved in the writing of the manuscript or the decision to publish the results.

**Ethics and Disclosure Statement:** Informed consent was obtained from all of the participants. The experiments and study were approved by the code of ethics of Brno University of Technology. The authors disclose no potential conflicts of interest, including any financial, personal, or other relationships with third parties that could inappropriately influence their work.

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
