# Peer review of "Human Psychophysiology Is Influenced by Low-Level Magnetic Fields: Solar Activity as the Cause"

_atmosphere, doi:10.3390/atmos12121600_

Round 1

Reviewer 1 Report

9.11.2021

Paper: Human Psychophysiology is influenced by Low-Level Magnetic Fields: Solar Activity as the Cause 

Authors:   Michael Hanzelka, JiÅ™í Dan, Pavel Fiala and PÅ™emysl Dohnal

The paper is interesting, but there are many doubts.

  1. Solar Activity (SA). You should add information about types of SA sources of data. Type of data? SSN, Solar wind, 10.7 cm, Dst and Kp ??? What is resolution of data?. E.G. You can take data from:  https://omniweb.gsfc.nasa.gov/
  2. Table 1. Small/pure statistic, in Table 1 is total numbers of respondents, only 49 ?
  3. Figure 1. The solar activity during the research. Please add information , what is on the axis OY . 2000??? add units . What about error bars ? what is 210 ???? is not a sum ???
  4. Figure 2 Solar activity correlated, is Pearson correlation coefficient, or another one ?
  5. Add some formulas, what do you calculated ?
  6. Figure 3 what is Slunce ??? use English !!! do not use Slovakian language in paper. Add more details  
  7. Next Figures up to Fig. 8 please add more details, clarify format !!!
  8. Table 2. The predicted number of sunspots and radio flux values with expected ranges (2019). 225 Please check Source: [22] sth is wrong
  9. Generally add more anabasis before conclusions
  10. Many links is out of order :   g.

  • https://doi.org/10.1016/S0273-1177(97)01097-1
  • https://thoughttechnology.com/pdf/manuals/SA7913%20V6.0%20BioGraph%20Infiniti%20Getting%20Started.pdf
  • https://research.stlouisfed.org/fred2/series/DJIA/downloaddata

Author Response

Thank you very much for your comments. Below is a comment on the individual comments. 

1) Solar Activity (SA). You should add information about types of SA sources of data. Type of data? SSN, Solar wind, 10.7 cm, Dst and Kp ??? What is resolution of data?. E.G. You can take data from:  https://omniweb.gsfc.nasa.gov/

MH: Added on the line: 79 - 83

2) Table 1. Small/pure statistic, in Table 1 is total numbers of respondents, only 49 ?

MH: Added on the line: 70 - 74

3) Figure 1. The solar activity during the research. Please add information , what is on the axis OY . 2000??? add units . What about error bars ? what is 210 ???? is not a sum ???

MH: Added on the line: 74

4) Figure 2 Solar activity correlated, is Pearson correlation coefficient, or another one ?

MH: Pearson correlation coefficient

5) Add some formulas, what do you calculated ?

MH: Added on the line: 152 - 158

6) Figure 3 what is Slunce ??? use English !!! do not use Slovakian language in paper. Add more details  

MH: Fixed in the Figure 3, 4, 5

7) Next Figures up to Fig. 8 please add more details, clarify format !!!

MH: Added on the Figure 8

8) Table 2. The predicted number of sunspots and radio flux values with expected ranges (2019). 225 Please check Source: [22] sth is wrong

MH: Added on the line: 257- 259

9) Generally add more anabasis before conclusions

MH: Added on the line: 260 - 264

10 Many links is out of order :   g.

MH: Links are tested and work.

Thank you for your kindness and attention.

Michael Hanzelka

Reviewer 2 Report

Manuscript ID: atmosphere-1465114

Title: Human Psychophysiology is Influenced by Low-Level Magnetic Fields: Solar Activity as the Cause

------------------------------------------------------------------------------------

Reviewer Report

The paper “Human Psychophysiology is Influenced by Low-Level Magnetic Fields: Solar Activity as the Cause” shows the evaluation of the impact of changes in solar activity on three significant human psychophysiological parameters: skin conductance, Electromyography.

The paper is well written and is highly important for many researchers, from physics, Psychophysiology. I have carefully read the paper which are closely described in the following two references:

Hanzelka, et al. Experiments with sensing and evaluation of ionosphere changes and their impact on the human organism, Measurement, 2015,10th, 2015. 299

Hanzelka, et al., Methods and Experiments for Sensing Variations in Solar Activity and Defining Their Impact on Heart Variability. Sensors 2021, 21, 4817.

Therefore, in my opinion, the paper can be published as it is.

Author Response

Thank you very much for recommending the article for publication.

Michael Hanzelka